# Invisible ship tracks show large cloud sensitivity to aerosol

Peter Manshausen[1✉], Duncan Watson-Parris[1], Matthew W. Christensen[2], Jukka-Pekka Jalkanen[3] & Philip Stier[1]

Cloud reflectivity is sensitive to atmospheric aerosol concentrations because aerosols provide the condensation nuclei on which water condenses[1]. Increased aerosol concentrations due to human activity affect droplet number concentration, liquid water and cloud fraction[2], but these changes are subject to large uncertainties[3]. Ship tracks, long lines of polluted clouds that are visible in satellite images, are one of the main tools for quantifying aerosol–cloud interactions[4]. However, only a small fraction of the clouds polluted by shipping show ship tracks[5,6]. Here we show that even when no ship tracks are visible in satellite images, aerosol emissions change cloud properties substantially. We develop a new method to quantify the effect of shipping on all clouds, showing a cloud droplet number increase and a more positive liquid water response when there are no visible tracks. We directly detect shipping-induced cloud property changes in the trade cumulus regions of the Atlantic, which are known to display almost no visible tracks. Our results indicate that previous studies of ship tracks were suffering from selection biases by focusing only on visible tracks from satellite imagery. The strong liquid water path response we find translates to a larger aerosol cooling effect on the climate, potentially masking a higher climate sensitivity than observed temperature trends would otherwise suggest.

Global warming caused by greenhouse gases has been partially masked by anthropogenic aerosols. The latest report of the Intergovernmental Panel on Climate Change's Working Group I[3] estimates the combined forcing of aerosols to be $-1.3 \pm 0.7$ W m$^{-2}$ (90% confidence range). This cooling results partly from the reflection of solar radiation by aerosol particles (direct effect) and to a larger part from changes to cloud properties (indirect effect). While the direct effect can be better constrained[7], aerosol–cloud interactions contribute the largest uncertainty of anthropogenic climate forcing[3,8].

Aerosol–cloud interactions stem from aerosols acting as cloud condensation nuclei (CCN), on which water condenses in supersaturated air. An increase in CCN causes an increase in cloud droplet number ($N_d$) if the total amount of liquid water does not decrease, brightening clouds[1]. But the liquid water path (LWP; a strong control on albedo) can change in response to a change in droplet number. The reduced size of the droplets lowers the speed at which droplets grow by collision coalescence, and can suppress precipitation, increasing both LWP and the area covered by clouds (cloud fraction, CF) in polluted regions[2]. By changing the entrainment and mixing processes with the surrounding air, increased $N_d$ can also lower LWP[9,10]. It has been suggested from the cloud response to a volcanic eruption[11] that the overall effect of aerosols on LWP is weak. Toll et al.[12] argue, on the basis of an analysis of shipping and industry emission effects, that this is because decreases and increases in LWP almost balance out. They suggest that climate models that show an increase in LWP with aerosol concentration suffer from poor parameterization and overestimate aerosol–cloud

cooling. As global climate models do not explicitly resolve shallow convection, the LWP response they predict has a strong dependence on parameterization[13].

So-called opportunistic experiments, where the cause of the aerosol perturbation is known and localized in time and space, have recently been widely used as an approach to quantify aerosol–cloud interactions[4]. The presence of a known emission source allows for a direct comparison of polluted and unpolluted clouds, excluding possible confounders.

Ship tracks, linear cloud features of increased reflectivity and cloud cover over the global oceans, are the most prominent example of such opportunistic experiments. They were first reported by Conover[14] and are attributed to the aerosol emissions of ships, mainly sulfate precursors and black carbon[6]. They have been studied extensively and shown to constrain aerosol–cloud interactions[5,6,15,16]. Previous ship track studies have provided evidence for negative LWP responses to aerosol[12,17]. If this negative response is large enough it can lead to the polluted cloud becoming less reflective, the opposite of the expected aerosol effect[18].

However, the use of ship tracks to quantify aerosol–cloud interactions may introduce selection biases, either by not allowing enough time for LWP reductions to occur via entrainment[19], or by selecting for shallow boundary layers (<800 m). Large eddy simulations show that, in deep boundary layers, the microphysical response to an aerosol source is hidden by natural variability, but is detectable by averaging along the emission trajectory[20]. Comparing entire regions of high and low shipping emissions[16,21] partly addresses biases, but this approach is

[1]Atmospheric, Oceanic and Planetary Physics, Department of Physics, University of Oxford, Oxford, UK. [2]Pacific Northwest National Laboratory, Richland, WA, USA. [3]Finnish Meteorological Institute, Helsinki, Finland. ✉e-mail: peter.manshausen@physics.ox.ac.uk

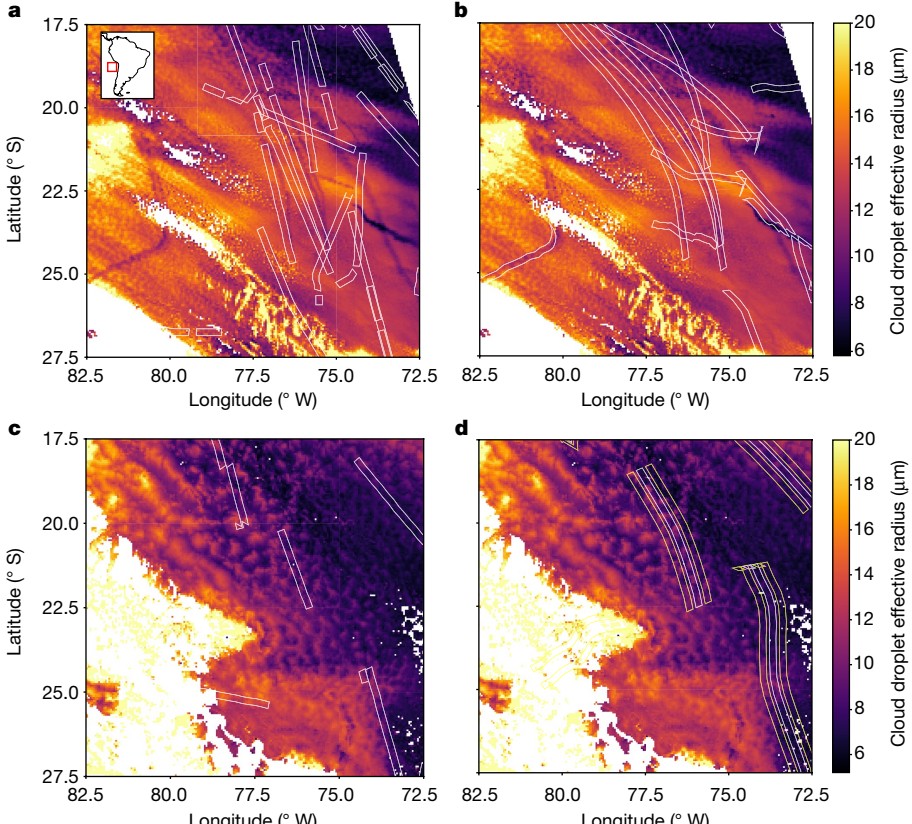

**Fig. 1 | A view of the Chilean Sc deck from the MODIS[41] satellite instrument.** The colour scale indicates cloud droplet effective radius. **a**,**b**, Grey boxes show the location of emissions (**a**) and where they are advected to with the wind (**b**). There are visible ship tracks in darker colours, most of which are well collocated to advected ship emissions. MODIS Aqua on 6 August 2018. **c**, The same region but for a day without visible ship tracks, even though there are ships. **d**, Advected emission locations and in addition the counterfactual retrieval boxes to either side of them (yellow boxes). MODIS Aqua on 14 September 2019, selected ships. Map of the retrieval location from cartopy[42].

confined to small regions. Recent work by Gryspeerdt et al.[22] uses ship positions and emissions data to convert the distance along the ship track to time since the emission of cloud-perturbing aerosol, showing a time-resolved picture of aerosol–cloud interactions. However, open questions remain about the justification for generalizing from ship track studies to global sensitivities.

In this study, we show that relying on visible ship tracks introduces a large selection bias. We propose a method that takes into account emissions that do not cause visible tracks. High-resolution ship emissions are used to establish where clouds were influenced by aerosols and to compare these locations to locations without ship emissions. The analysis does not rely on hand-logged ship track locations, only on shipping emission data. We thereby remove the selection bias for visible track formation. This leads to larger positive LWP responses to the aerosols emitted by ships.

## Finding invisible ship tracks

Previous work has largely relied on the identification of ship tracks by eye from near-infrared satellite imagery (for example, 2.1 or 3.7 μm channels) and the logging of their positions by hand[15,22,23]. Here, we use datasets of ship emissions, which are advected with the wind using the HYSPLIT model[24] (Methods) until the time of the overpass of the Moderate Resolution Imaging Spectroradiometer (MODIS) on the satellites Aqua and Terra. The satellite-retrieved cloud properties at these locations are the ones subject to aerosol pollution by ships. We compare these to clouds in unpolluted locations nearby, such that the meteorological conditions are similar. If a ship track is visible, the in-track and out-of-track properties are retrieved, just like in the ship track studies mentioned earlier that involve logging positions by hand. Crucially, retrieval of polluted and unpolluted cloud properties also happens in the more common case of ship emissions that do not produce visible tracks.

Figure 1 exemplifies our approach: the framed retrieval sections are estimates of where aerosol has been carried by the wind between the times of their emission and the satellite retrieval. Three cases can be distinguished: (1) a ship track is visible and it lies within the retrieval section; (2) no ship track is visible but there is a retrieval section; and (3) a ship track is visible but (partly) outside of any retrieval section. Some of the scenes falling into cases (2) and (3) will arise because of imperfect collocation of advected emissions with satellite retrievals. However, case (2) scenes would arise even with perfect collocation. Only a fraction of shipping emissions actually form visible ship tracks[21]. This becomes very clear on days when there are no visible ship tracks at all, as shown in Fig. 1c,d. Scenes that fall into case (3) can stem from imperfect collocation or ships missing from the emissions dataset.

## Visible versus invisible tracks

The effect of ship emissions on cloud properties can be detected even if no tracks are visible by eye in an effective radius retrieval. To show this, we focus on the region of the Chilean stratocumulus (Sc) decks known to display visible ship tracks. We separate days by eye when there are any visible ship tracks in the region of interest, as in Fig. 1a,b, from those days when there are none, as in Fig. 1c,d.

Figure 2a,b show a comparison of cloud properties on days with visible tracks and days with no visible tracks. Plotted are the ratios of

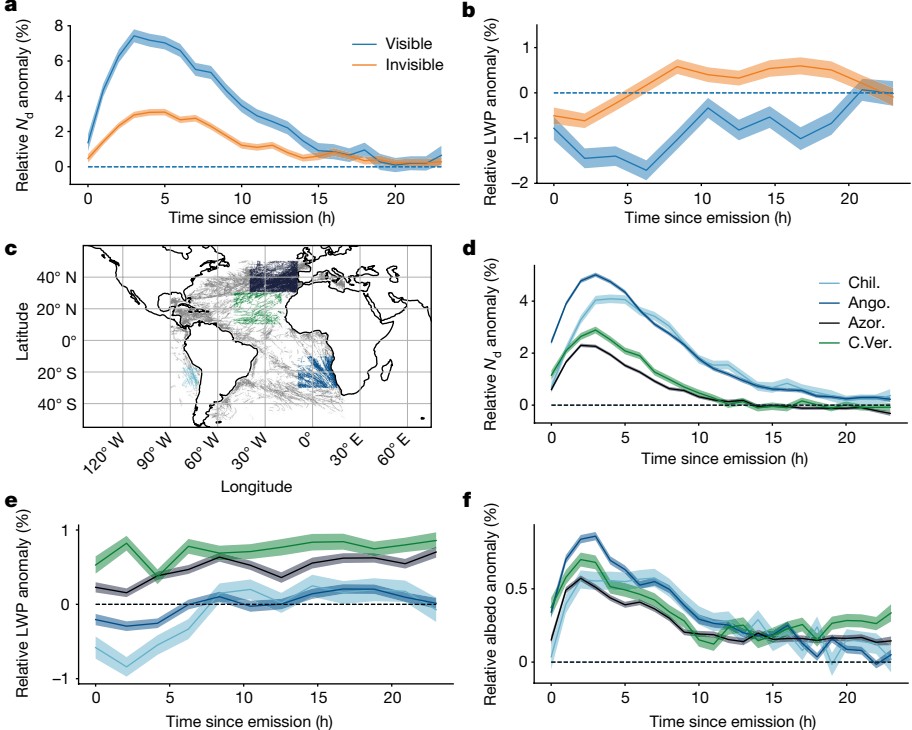

**Fig. 2 | There are cloud property changes even when there are no visible tracks. a**,**b**, Comparing $N_d$ (**a**) and LWP (**b**) in the Chilean Sc between those days when ship tracks are visible (blue) and those days when none are visible (orange) in the region. Plotted are ratios of hourly means of in-track and out-of-track properties, so that a value larger than 0% means an enhancement in the track. Shaded areas show standard errors of the means. LWP anomalies are given in 2 h means rather than 1 h because of noisier data. Retrievals from the MODIS-cloud product on Aqua and Terra[41], all for 2014–2019, 25% of days with visible tracks. Also, there are measurable responses to ship emissions everywhere. **c**, A plot of a comparison of in-track and out-of-track properties in four regions: the Chilean (Chil., light blue) and Angolan/Namibian (Ango., blue) Sc, and the Cu around the Azores (Azor., navy) and Cabo Verde (C.Ver., green). The regions are shown in different colours representing the ship track location data of around six days. **d**–**f**, Plots showing relative $N_d$ (**d**), LWP (**e**) and albedo (**f**) anomalies. Shaded areas show standard errors of the means. Retrievals from the MODIS-cloud product on Aqua and Terra[41], all for 2014–2019. Map of the different regions from cartopy[42].

in-track over out-of-track hourly averaged $N_d$ (Fig. 1a) and LWP (Fig. 1b) retrievals. There is a significant enhancement in $N_d$ even on days when no ship tracks can be seen by eye. This surprising result could indicate 'blurred' tracks, which cannot easily be seen by eye. These diffuse changes could become apparent when averaging over the length of the track[20] or over many tracks, as we have done here (about 600 are needed, see section 'Radiative forcing and detectability'). Ship tracks may also be invisible in the effective radius retrieval but visible in droplet number. This can occur when the droplet number increases together with the liquid water content, as shown in figure S1 of Gryspeerdt et al.[22].

The time evolution of increases in $N_d$ is similar in the visible and invisible cases, peaking at 3–4 h and persisting for up to 20 h after the emission of aerosol. Contrary to the $N_d$ response, the LWP response is qualitatively different between the visible and the invisible cases. Whereas, in agreement with results from Toll et al.[12], the response in the visible case is weakly negative, it turns weakly positive in the invisible case. For the resulting sensitivity of LWP to $N_d$ perturbations, defined in equation (1), this means a negative sensitivity for visible tracks and a positive sensitivity for invisible tracks. The time development is plotted in Extended Data Fig. 1. In agreement with Glassmeier et al.[19], the magnitude of sensitivity increases. Here, however, this seems to be due to the decreasing $N_d$ anomaly with time rather than an increasing LWP anomaly.

To answer the question of why some tracks are invisible, we study the background meteorological state. Extended Data Fig. 2 shows meteorological conditions from ERA5 reanalysis data for 'invisible' and 'visible' retrievals. Visible ship tracks occur in cloud decks that are capped by stronger inversions (as quantified by estimated inversion strength, EIS), are shallower, lie above colder sea surfaces and higher relative humidity conditions, and show lower background $N_d$. These findings are consistent with those obtained over the North Pacific by Durkee et al.[25].

## Invisible ship tracks everywhere

Invisible ship tracks can also be detected in regions that have not been widely studied for ship tracks previously. One such area is the East Atlantic around Cabo Verde; in addition, we study the Northeast Atlantic around the Azores. Both of these areas tend to have lower cloud fractions and the low clouds are mostly shallow cumulus (Cu). These two regions, and the Chilean and Namibian Sc, are shown in Fig. 2c. All regions show similar relative $N_d$ anomalies and time evolutions (Fig. 2d). However, LWP responses (Fig. 2e) are markedly different. Importantly, the region that was not previously studied, the trade Cu region of Cabo Verde, shows the largest LWP enhancement. In all cases, there is little time development of LWP. Albedo enhancement (Fig. 2f) follows a similar pattern as $N_d$. Strikingly, the stronger LWP response seems to make up for the smaller $N_d$ enhancement, such that the Cabo Verde region, which displays almost no visible tracks[5], has a very similar albedo enhancement to the Chilean Sc. Previous studies of trade Cu at the shipping lane or regional scale[21,26,27] have not found statistically significant anomalies, with the exception of $N_d$ increases[21].

The pattern of strong LWP responses in the trade Cu regions is confirmed in Fig. 3. Plotted are the ratios of regional means of in-track and

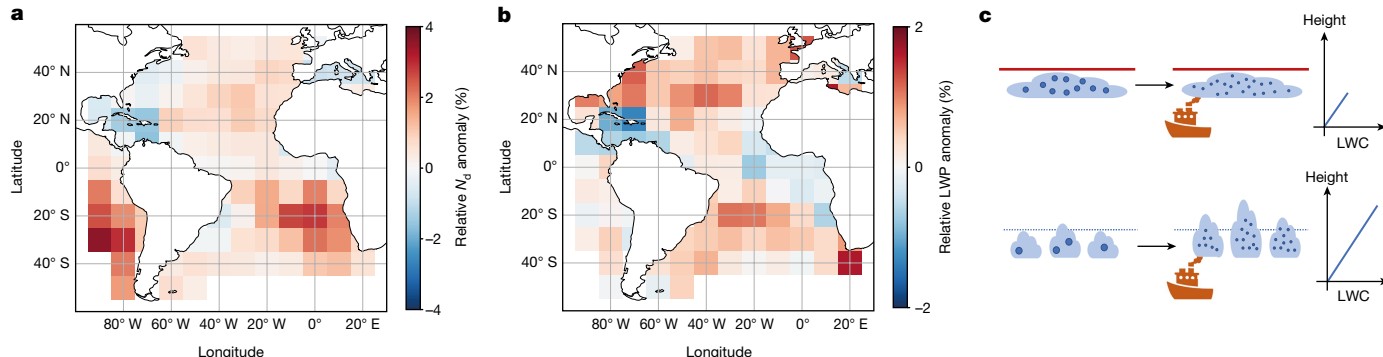

**Fig. 3 | More droplets in Sc, more LWP in Cu. a,b,** Heatmap of relative $N_d$ (**a**) and LWP anomaly (**b**), that is, mean of in-track properties divided by mean of out-of-track properties. $N_d$ responses are strongest in the Sc regions, but LWP responses are strongest in the trade Cu. Retrievals from MODIS-cloud product on Aqua and Terra[41], all for 2014–2019. Map from cartopy[42]. **c,** The possible mechanism for a more positive LWP response in weak-inversion regions. Previous ship track studies focused on regions with stable conditions and relatively high background $N_d$ (top). There, cloud depth cannot increase because of the strong inversion topping the cloud. Invisible tracks occur more often in unstable, low background $N_d$ clouds (bottom). When more CCN are provided by a ship, these clouds can deepen. As liquid water content (LWC) increases linearly with height, LWP, its integral, increases quadratically.

out-of-track properties. The most positive response in LWP appears in the trade wind regions on either side of the equator, whereas the response in $N_d$ (implicitly averaged over all times since emission) is strongest in the Sc regions of Namibia and Chile.

Higher LWP sensitivity in the trades correlates with weak or negative inversions and low background $N_d$, shown in Extended Data Fig. 3. Both help to explain a larger LWP enhancement in these regions: the $N_d$–LWP relationship is known to be non-linear, with an increase in LWP in the low-$N_d$ regime and a decrease in the high-$N_d$ regime[28].

The LWP increase is due to suppression of precipitation, but for LWP to increase, clouds also need to deepen, because in adiabatic clouds LWP $\propto \Delta Z^2$, with the cloud depth $\Delta Z$ (refs. [29],[30]). However, cloud deepening upwards and an associated increase in LWP is only possible if the cloud is not topped by a strong inversion (Fig. 3c). Extended Data Fig. 4 shows the anomaly in cloud top temperature, which decreases with cloud height. Regions with stronger LWP responses seem to show cloud top increases (temperature decreases). However, the signal is not strong enough to enable a significant time-resolved study. At the same time, changes in adiabaticity or deepening downwards could also increase LWP. In Extended Data Fig. 7, anomalies in $N_d$ and LWP are shown as a function of stability, with the data separated into four bins by their EIS. The more stable the environment, the larger the $N_d$ anomaly, with no enhancement in very unstable conditions. The LWP response is more positive in the two middle quartiles, whereas the enhancement is weak in stable conditions, again with no enhancement in very unstable conditions. This could be attributed to a non-linear effect of aerosol on LWP, but is more likely to be linked to the fact that there is no observed effect on $N_d$. Lower EIS is correlated with deeper boundary layers, so the increase in LWP in low-EIS regions is contrary to results from visible track observations by Toll et al.[31] as well as climatological observations by Possner et al.[32], who find a more negative LWP response in deeper boundary layers.

Together, these results imply that the same unstable conditions, which could be responsible for tracks being invisible (blurred), could also cause the LWP response to be more positive. The large positive LWP responses in relatively unstable conditions can be seen as a generalization of the findings of Christensen and Stephens[15]: the authors observe LWP increases and cloud deepening for ship tracks in open cell Sc, whereas there was a small negative LWP response in closed cells. This matches modelling results, showing that in order to maximize cloud water, there are 'optimal concentrations' of aerosol, with a higher optimal value (that is, more aerosol-limited clouds) in unstable conditions[33].

A large eddy simulation study of trade Cu clouds by Seifert et al.[34] found cloud deepening with increasing $N_d$, but only small LWP increases. However, whereas Seifert et al. focus on the equilibrium states of their simulations, Spill et al.[35] find a more robust positive LWP response and cloud deepening in simulations of more realistic, transient, thermodynamic conditions. Yamaguchi et al.[36] also find deepening and LWP increases in trade cumulus, but only conditionally with no-wind-shear conditions. A similar LWP response has also been reported in the warm base of deep convective clouds[37].

These findings suggest that ship track studies that have focused exclusively on the high-$N_d$, high-inversion-strength regions have missed the large sensitivity of clouds to aerosols in low-$N_d$, low-inversion-strength regions.

## Radiative forcing and detectability

Whereas LWP adjustments were previously shown to contribute a small but probably positive forcing[8,12], our results indicate the potential for a substantial negative forcing. Following Bellouin et al.[8], and extrapolating to the rest of the globe (Methods), we estimate the forcing from LWP adjustment (and its 90% confidence intervals) to be −0.76 (−1.03, −0.49) W m$^{-2}$. This is of opposite sign and larger magnitude than the estimate of 0.2 ± 0.2 W m$^{-2}$ given in the latest Intergovernmental Panel on Climate Change report[3] (section 7.3.3.2.1). This substantially larger estimate is mostly due to relatively strong responses in LWP in unstable atmospheric conditions, where the $N_d$ response is weak, resulting in a large sensitivity.

Our approach also demonstrates a viable technique for monitoring experiments of marine cloud brightening (MCB), the targeted emission of aerosols into ocean clouds to increase reflectivity and thereby mask global warming. Seidel et al.[38], and more recently Diamond et al.[21], have argued that the effect of MCB would only be detectable with remote sensing on a regional scale after several years of emissions. However, knowing the positions of emissions enables detection of their effects with as little as one month of observations. Extended Data Fig. 5 shows that $N_d$ anomalies can be confidently detected with 30 days of data, or equivalently about 600 individual ships. To confidently detect albedo changes, four months of data are necessary. For the purpose of MCB experiments, there is the caveat that we do not consider cloud fraction changes or possible decreases in cloudiness at larger distances from the tracks, as found by Wang et al.[39]. Furthermore, our albedo measurements rely on the proxy of cloud optical thickness, similarly to other studies of this kind[15,23].

## Discussion and conclusion

Based on the analysis of more than two million ship paths across the Atlantic over six years, we have been able to show that the effect of ship emissions can be directly observed even if they do not form visible ship tracks. These 'invisible' ship tracks have markedly different properties than visible ship tracks, in particular a positive LWP response even in probably non-precipitating clouds. Similarly, the trade wind regions not previously studied for ship tracks (rather than regional shipping effects[21,26,27]) show the strongest positive LWP response. Together, this shows that previous studies using ship tracks as general proxy for aerosol–cloud interactions have been subject to a strong sampling bias. Relying on visually identifiable ship tracks confines the analysis to specific meteorological conditions and small geographical regions where visible ship tracks occur. We hypothesize that the strong inversion found in these regions in particular limits LWP enhancement because clouds cannot deepen here. At the same time, the strong inversion could be necessary to form tracks that are clearly visible, rather than blurred or diffuse. This contradicts the assumption that findings from visible tracks generalize to other conditions and regions[11,12,31]. It also links to the importance of accurate descriptions of shallow convection, which are currently not explicitly represented in global climate models. Nevertheless, our measurements agree more closely with the models than with previous observations. Our results imply a substantially larger negative radiative effect (and associated cooling) from LWP adjustments than previously reported.

Our method may suffer from imperfect collocation of emissions and satellite retrievals, leading to some in-track and out-of-track properties actually belonging to the other category. The origin of this is the inherent uncertainty of the reanalysis data used to advect the track, and uncertainties related to trajectory modelling and the original emission location. This effect gets stronger the more time has elapsed since emission, as the advection of emissions becomes more uncertain with time. However, all results are presented as departures from the ground state. This means that in a hypothetical situation of large collocation errors we would expect no anomalies.

Retrieval uncertainties also need to be considered: LWP retrievals in the broken cloud scenes of the trade wind Cu are much more uncertain than in Sc decks[40]. Cloud top temperature retrievals suffer from uncertainties under strong-inversion conditions. Biomass-burning smoke and Sahara dust can interfere with retrievals in some parts of the Atlantic. However, in-track and background measurements will be equally affected by retrieval errors. Therefore, we do not expect the results to be strongly impacted.

Splitting the data in the traditional ship track regions into days with visible tracks and days without highlights how taking into account only the former can lead to a bias even within a region. This is only a first approximation of the differences between individual visible and invisible tracks, and may introduce a degree of subjectivity. However, for a consistent estimate of sensitivities and ultimately radiative forcing, all tracks need to be taken into account, and the distinction between visible and invisible tracks becomes irrelevant.

Although aspects of the radiative forcing owing to aerosol–cloud interactions remain uncertain, this work is a meaningful step towards observational constraints across all cloud regimes. In particular, it highlights the importance of using lines of evidence that are representative of how aerosol sensitivities change with meteorological conditions.

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

## Methods

### Scope

Geographically, this study considers the larger part of the Atlantic between (50 °S, 50 °N) and (−90 °W, 20 °E) as well as the Sc deck in the Southeast Pacific off the Chilean coast. The size of the region is limited by computational cost and we chose to place it so it would cover two Sc regions. Our data are for the year 2019 and, in the case of the Chilean Sc region of (−35 °S, 0 °N) by (−90 °W, −70 °W), in addition for 2018. In Fig. 3, we chose windows of (−30 °S, −17 °S) by (−82.5 °W, −72.5° W) for the Chilean Sc, (30 °N, 50 °N) by (−40 °W, −10 °W) for the Azores Cu, (10 °N, 30 °N) by (−50 °W, −20 °W) for the Cabo Verde Cu, and (−30 °S, −10 °S) by (−10 °W, 15 °E) for the Namibian Sc. The Sc positions were chosen for highest cloud fraction, whereas the Cu regions were chosen for overlap with (future) in-situ measurement campaigns in the North Atlantic such as the ACRUISE field campaign.

### Datasets

Ship emissions were calculated using the model of Jalkanen et al.[43] from automatic identification system data. These data, which contain $SO_x$ emissions, are at hourly and 0.05 ° resolution. The resolution is limited by the studied area; however, a change from 0.01 to 0.05 ° does not affect results as further steps of the pipeline have higher spatial uncertainty. While previous studies are limited by the number of tracks they can consider, of the order of thousands [12,22], we use of the order of two million equivalent ship paths. Note that although we calculate emission magnitudes, the analysis in this study uses only the emission locations. Even though there are uncertainties in the emission amounts, these do not alter the findings of the study. To transport the emissions, the HYSPLIT advection scheme (see next section) uses ERA5 reanalysis data of wind speeds at 3-h intervals and 0.25 ° resolution[44]. Satellite retrievals of cloud properties (cloud droplet effective radius $r_{eff}$, optical depth, LWP) are from the Level-2 cloud product MOD 06 of the MODIS on the Aqua and Terra satellites[41], collection 6.1. Only properties of liquid clouds are used, as given by the MODIS-cloud property mask. $N_d$ is calculated, following Quaas et al.[45], from effective radius retrievals and cloud optical thickness (see also the derivation to equation (11) in Grosvenor et al.[46]). Albedo is calculated, as in Segrin et al.[23], from cloud optical thickness.

### Advection

Ship positions are detected using the Trackpy library[47]. The HYSPLIT model[24] advects these positions for 24 h, using ERA5 winds. Note that ERA5 data come at 3-h intervals and 0.25 ° resolution, but that the HYSPLIT model interpolates for a more exact estimate. Settings of 20 m initial height and the 'input model data' mode for the vertical motion method are used. The HYSPLIT functionality for trajectories, rather than that for dispersion, is used here, so as to find the advected emission location. This is to limit computational cost, as the final datasets analysed are large. Each satellite retrieval is then collocated with the emissions that occurred in the 24 h before the retrieval, advected to the positions at retrieval time. As the emissions data time resolution is only hourly, it is interpolated to a 5-minute period to not miss parts of the track.

### Counterfactuals

We need to compare the polluted clouds to the counterfactual situation in which they would not have been polluted. As ship emissions are strongly localized, we can assume that the region a small distance away will not have been polluted and can be used as a proxy for the counterfactual. Choosing the distance from the pollution source at which the counterfactual is retrieved is a trade-off: the smaller the distance, the more similar the retrieved cloud is to the counterfactual, but it also risks being influenced by pollution. This is particularly true when the collocation is more uncertain, that is, at times further away from

emission. In this study, cloud properties within a 10 km radius around the pollution are retrieved for the in-track properties. Out-of-track properties are retrieved at a 30 km distance, also with a 10 km radius. Out-of-track properties are always retrieved on both sides of the track and averaged. Some error may be introduced by strong background aerosol gradients, especially when the distributions are non-linear. Then, the counterfactual constructed by averaging the areas to either side may be an overestimate (underestimate) for a convex (concave) function in the across-track direction. This explains the unphysical $N_d$ decrease in tracks, for example, in the Caribbean. Over large enough areas, backgrounds should, on average, be neither convex nor concave, so when averaging over the entire observation region, these effects should be negligible.

### Distinction between 'visible' and 'invisible' ship track days

In the case of the Chilean Sc, we distinguish days when there are visible tracks (quasilinear cloud features with a discernible head) on a given day from those when there are no visible tracks, as described in the section 'Visible versus invisible tracks'. This ensures high confidence in the retrievals labelled 'invisible'. In the 'visible' category, some retrievals may not actually be related to visible ship tracks. This can be for different reasons such as local meteorology, low emission amount or imperfect collocation. Taking this 'invisible'-conservative approach, we make sure that the invisible ship tracks we find could not have been retrieved using the conventional hand-logging approach, as on those days not a single ship track is visible. Nevertheless, this is only for the purposes of highlighting the historical bias, and our final determination of global LWP sensitivity does not depend on this distinction.

### Calculation of relative anomalies

The retrieved cloud properties often follow skewed distributions with strictly positive values and heavy tails at high values. This is exacerbated when taking ratio distributions. For example, the distribution of in-track-LWP divided by out-of-track-LWP has the surprising property of having a mean larger than 1, while the ratio distribution of the out-of-track-LWP divided by in-track-LWP also has a mean larger than 1. This incorrectly seems to indicate that average in-track-LWP is simultaneously larger and smaller than average out-of-track-LWP. It is, however, an artefact, resulting from the fact that the operations of taking the mean and taking the ratio of distributions do not commute (compare with Jensen's inequality[48]). To avoid misleading results, relative anomalies (for $N_d$, LWP and albedo) are reported as ratios of means rather than as means of ratios. Means are either taken over time windows, as in Fig. 2, over regions, as in Fig. 3, or over all of the data, as in the forcing calculation.

### Radiative forcing

From the responses in $N_d$ and LWP, we estimate the sensitivity as

$$\beta_L = \frac{d \ln LWP}{d \ln N_d}. \tag{1}$$

This is done by separating the ship track observations by EIS into four bins of equal data points, following the argument discussed earlier about the dependence of sensitivities to inversion strength. This enables an extrapolation to unobserved regions to approximate the global radiative forcing. Bins, sensitivity values and uncertainties obtained by bootstrapping are given in Extended Data Table 1. For LWP only the observations at least 5 h after emission are taken into account, when the response plateaus. Likewise, for $N_d$ only the observations before 5 h are used, when the enhancement is strongest. Following the calculation of Bellouin et al.[8], the sensitivity can be used to estimate the rapid adjustment of LWP ($RA_{LWP}$) to aerosol:

$$RA_{LWP} = \triangle \ln N_d \, \beta_L S_L c_L, \tag{2}$$

# Article

where $\Delta \ln N_d$ is the fractional anthropogenic change in droplet number, $S_L$ is the sensitivity of top-of-atmosphere radiation to LWP, and $c_L$ is the effective cloud fraction for rapid adjustments in LWP. $c_L$ is calculated according to

$$c_L = \frac{\langle C_L \beta_L \, \Delta \ln N_d \, R_{SW} \rangle}{\langle \beta_L \rangle \langle \Delta \ln N_d \rangle \langle R_{SW} \rangle},\tag{3}$$

with the liquid cloud fraction ($C_L$) downwelling shortwave radiation ($R_{SW}$) and regional averaging indicated by angular brackets. $C_L$ is obtained from MODIS Terra and Aqua between 2014 and 2019, depicted in Extended Data Fig. 6a, with the average being 10% higher for the Terra midday overpass than for the Aqua afternoon one. Patterns of $\Delta \ln N_d$ are calculated as in Bellouin et al.[8] from $N_d$ sensitivity to aerosol optical depth (AOD) and anthropogenic AOD, for which two estimates are taken from Bellouin et al.[49] and Kinne[50]. The average of the latter is 20% lower than the former. The higher (lower) bound for $c_L$ is obtained by multiplying Terra (Aqua) $C_L$ with the Bellouin (Kinne) $\Delta \ln N_d$ pattern and including (excluding) land.

The approach takes into account the spatial covariability of cloud fraction, LWP sensitivity, anthropogenic droplet number change and radiation. The spatial pattern of $\beta_L$ is shown in Extended Data Fig. 6b. Strikingly, Sc regions show up with low sensitivities. $S_L$ values are taken from Bellouin et al.[8]. Combining the confidence ranges with the Monte Carlo method from Bellouin et al.[8] gives the medians and 90% confidence ranges reported. Increasing the number of EIS bins to five or six changes the value by less than 10%.

A plot of the estimated radiative forcing from LWP adjustments is given in Extended Data Fig. 6c. The Sc regions (where ship tracks have been studied previously) are among those with the smallest radiative forcing, whereas tropical regions show a stronger negative forcing, linked to large LWP sensitivities. Although the exact method of weighting calculated sensitivities seems to have only a small effect, the selection of data from which to calculate sensitivities in the first place is important for the resulting radiative forcing. As is apparent, for example, in Extended Data Fig. 7, the $N_d$ anomaly is strongly time dependent, reaching a maximum around 3 h after emission. Extended Data Table 1 therefore presents the sensitivities obtained by using the $N_d$ perturbation from the first 5 h after emission, around its maximum. The LWP anomaly is taken only from the time after 5 h, when it plateaus. Using the entirety of the data would lead to even higher sensitivity estimates and consequently more negative estimates of radiative forcing.

## Data availability

ERA5 data are freely available from https://cds.climate.copernicus.eu/. MODIS data are freely available from https://ladsweb.modaps.eosdis.nasa.gov/search/. Emission datasets were obtained from Jukka-Pekka Jalkanen (Jukka-Pekka.Jalkanen@fmi.fi). The complete collocated data for ship tracks studied here including cloud property measures like $N_d$ and LWP has been archived by CEDA under https://doi.org/10.5285/2d0f8bb3927b4f75ae75276705858f68. Source data are provided with this paper.

## Code availability

Code for the production of the collocated data has been archived under https://doi.org/10.5281/zenodo.6556425.

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

**Acknowledgements** This work was supported by the European Union's Horizon 2020 research and innovation programme under Marie Skłodowska-Curie grant agreement No. 860100 (iMIRACLI). Analysis was performed and data stored with infrastructure provided by the UK Centre for Environmental Data Analysis. D.W.P. and P.S. were supported by the UK Natural Environment Research Council project ACRUISE (NE/S005099/1). P.S. also acknowledges support from the European Research Council Project RECAP under the European Union's Horizon 2020 research and innovation programme grant No. 724602 and from the FORCeS project under the European Union's Horizon 2020 research and innovation programme grant No. 821205. J.-P.J. acknowledges the support of the European Union's Horizon 2020 research and innovation programme under grant agreement No. 874990 (EMERGE). M.W.C. acknowledges The Pacific Northwest National Laboratory (PNNL) is operated for DOE by Battelle Memorial Institute under contract DE-AC05-76RLO1830. We thank E. Gryspeerdt for helpful discussions on interpreting the large LWP response and for providing data on anthropogenic $N_d$ enhancements; J. Runge for helpful comments on causality; A. Douglas for helpful comments on the manuscript; and M. Yang for coordinating the ACRUISE project and fruitful scientific discussions. We gratefully acknowledge the NOAA Air Resources Laboratory for the provision of the HYSPLIT transport and dispersion model used in this publication. This work reflects only our view and the European Climate, Infrastructure and Environment Executive Agency is not responsible for any use that may be made of the information it contains.

**Author contributions** P.M. developed the concept of the study and designed its implementation together with P.S., D.W.-P. and M.W.C. J.-P.J. produced the ship emissions dataset. M.W.C. converted the ERA5 data for use with the HYSPLIT model and provided python code to run it. P.M. wrote the code for collocating the datasets and analysed the data. All authors contributed to the interpretation of the results. P.M. drafted the manuscript with contributions and review from all co-authors.

**Competing interests** The authors declare no competing interests.

**Additional information**
**Correspondence and requests for materials** should be addressed to Peter Manshausen.

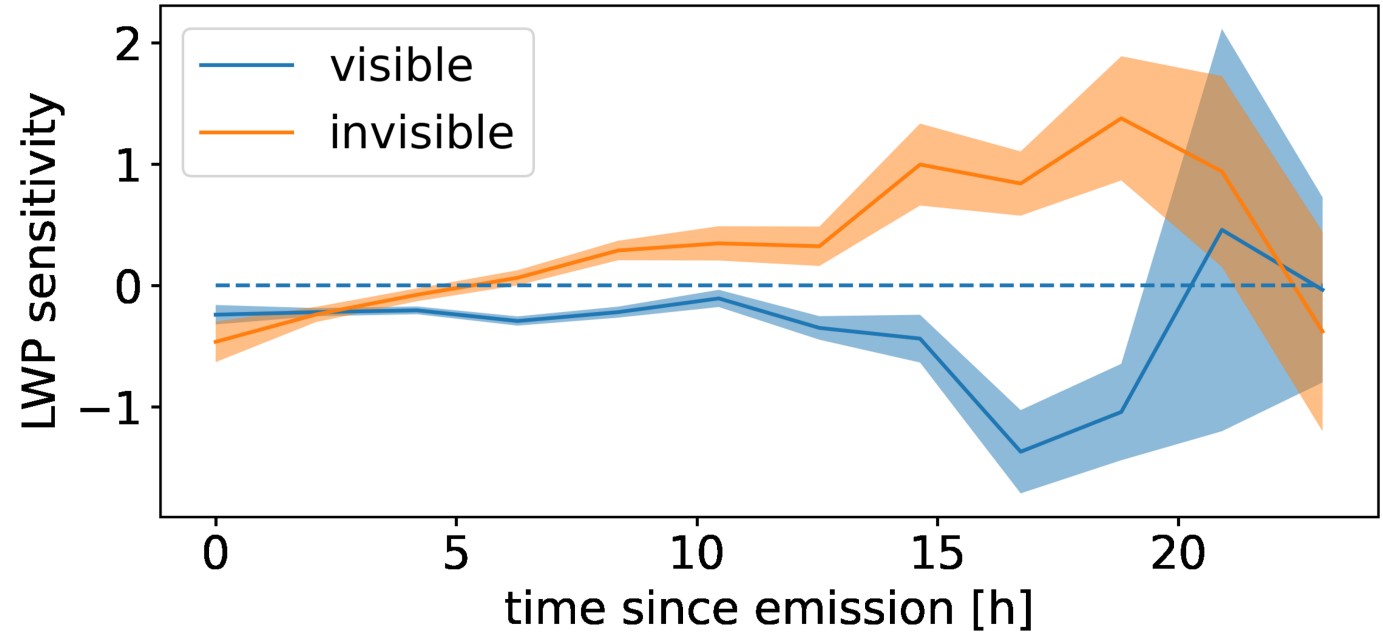

**Extended Data Fig. 1 | Time dependence of resulting sensitivity.** The time-resolved LWP sensitivity $\beta_L$ in the Chilean Sc on those days when ship tracks are visible (blue) and those when none are visible (orange) in the region. The data is the same as that presented in Fig. 2 (MODIS 2014-2019)[41], with the error propagated from the standard error of LWP.

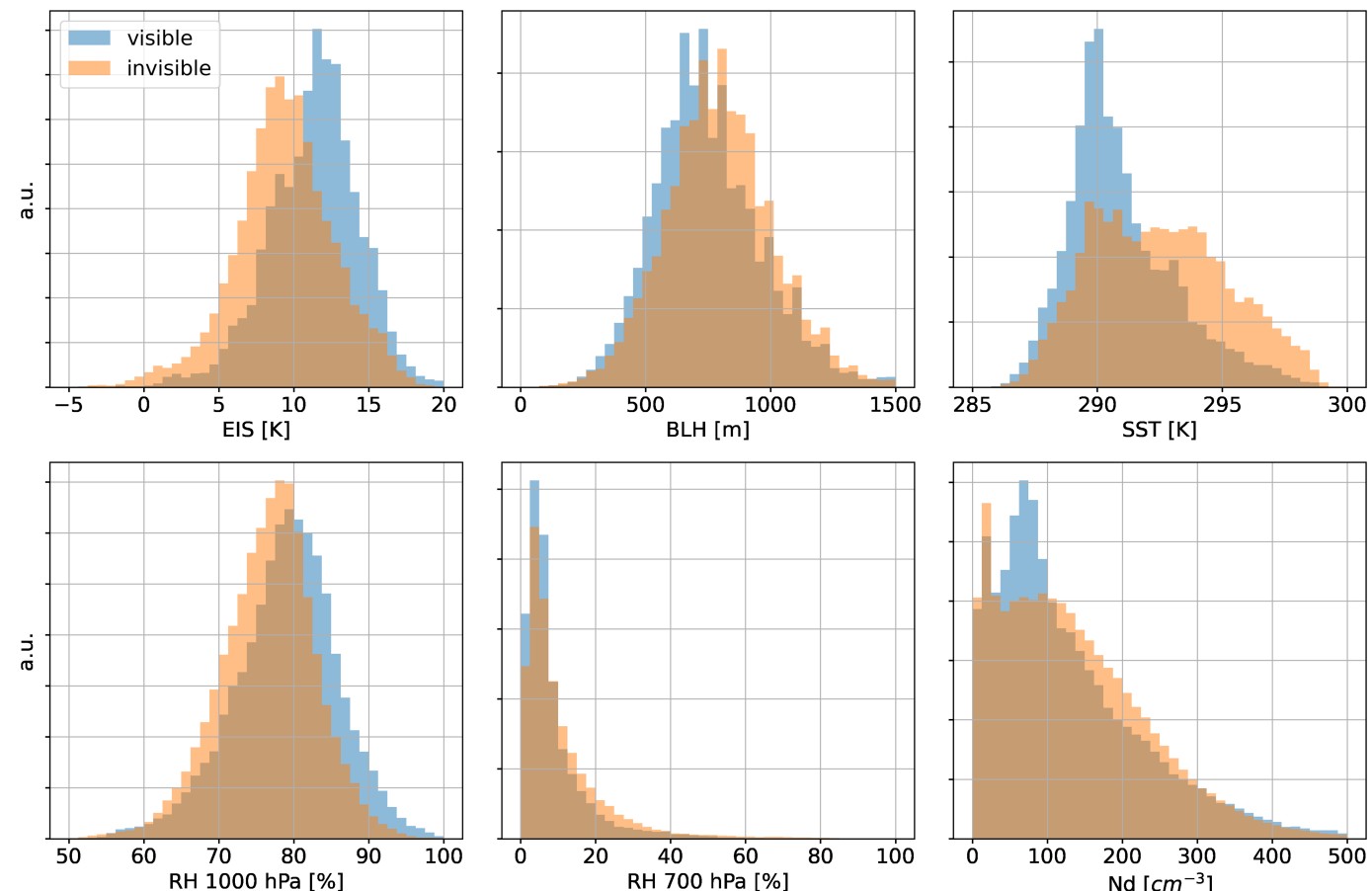

**Extended Data Fig. 2 | Visible and invisible tracks differ in their meteorological background.** Meteorological conditions of days that show any ship tracks and no ship tracks for the Chilean Sc in 2019. Plotted are density histograms in arbitrary units. Data is from ERA5[44], evaluated at the advected shipping emission locations and from the MODIS-cloud product on Aqua and Terra[41] for $N_d$.

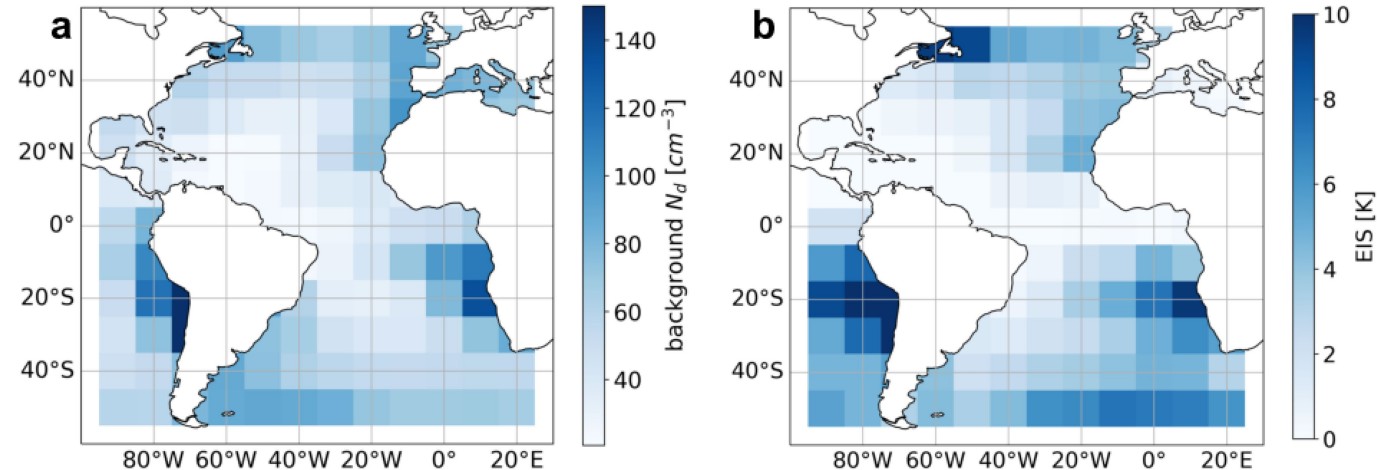

**Extended Data Fig. 3 | The responses to shipping emissions are correlated to stability and background $N_d$.** Plotted is a heatmap of inversion strength and background droplet number. Data is from ERA5[44] for EIS, evaluated at the advected shipping emission locations. The $N_d$ data is from the MODIS retrieval[41] to the sides of the emission locations. Maps from cartopy[42].

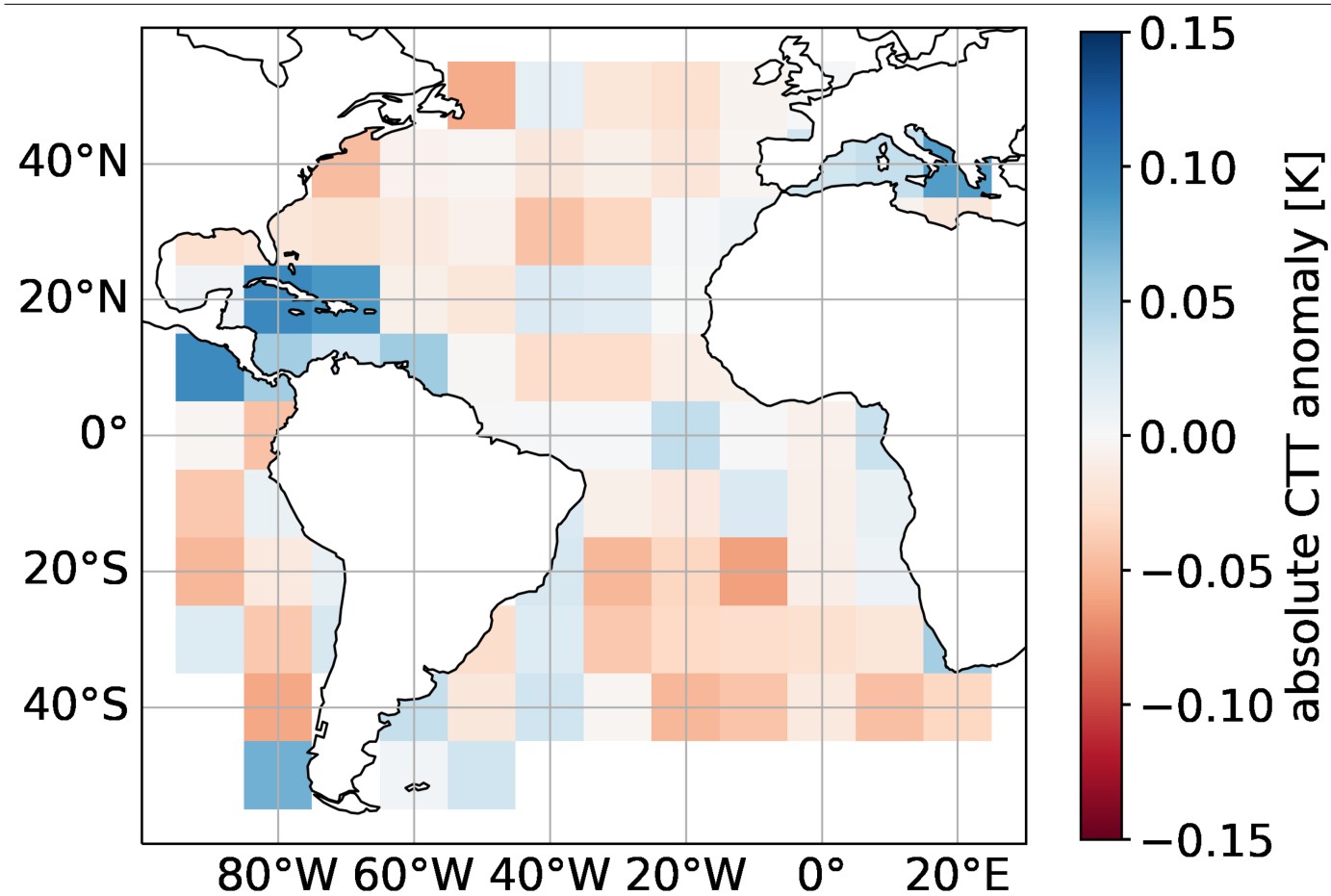

**Extended Data Fig. 4 | Stronger LWP changes may be due to cloud deepening, but data is noisy.** Heatmap of cloud top temperature (CTT) anomalies. Note that this heatmap is of absolute rather than relative differences like in Fig. 3, and that the color bar is reversed. Retrievals from MODIS-cloud product on Aqua and Terra[41], all of 2014-2019. Map from cartopy[42].

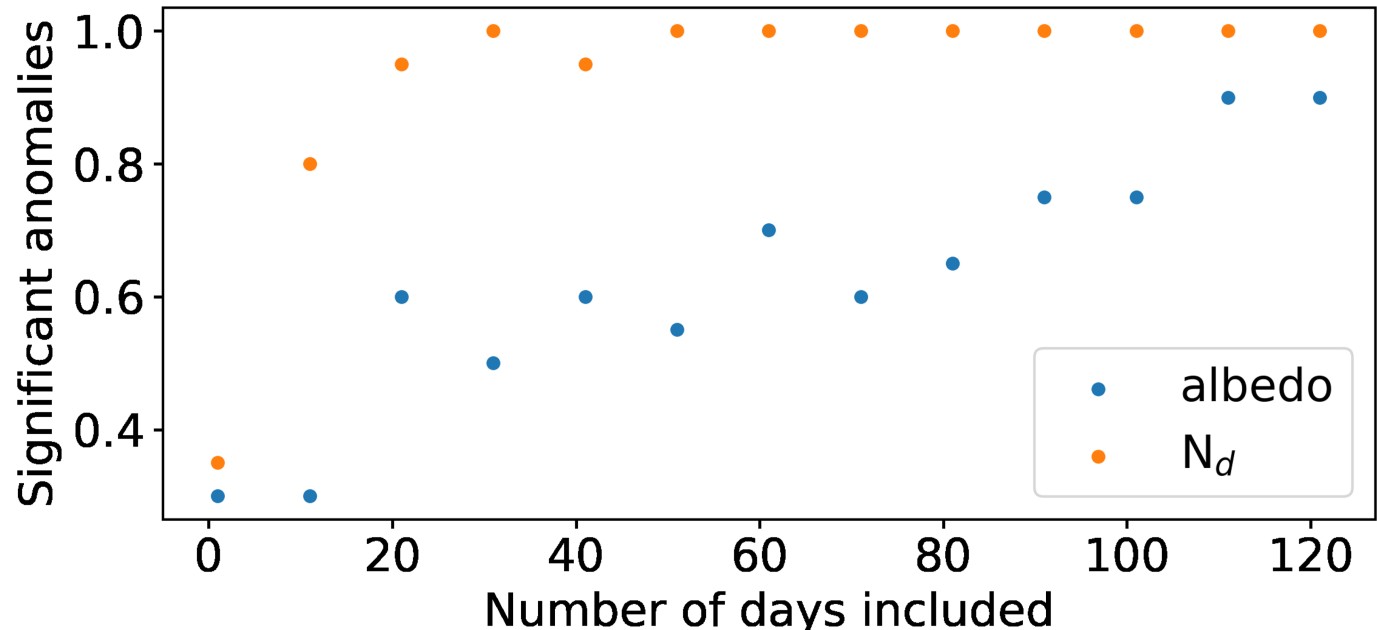

**Extended Data Fig. 5 | Changes to $N_d$ and albedo can be detected in a small region after only one and four months, respectively.** Shown is the fraction of significant anomalies for albedo and $N_d$ as a function of the number of days studied. This is for the Chilean Sc described above. For a given number of days n, 20 samples of n days are drawn from the total observation period and anomalies are calculated. This is to emulate a random draw of observation days. Anomalies are taken to be significant if they are larger than two standard errors, equivalent to a Student's t-test at 95% confidence. We assume that we are able to make a confident detection if in 95% of draws, so 19 out of 20, we observe significant anomalies.

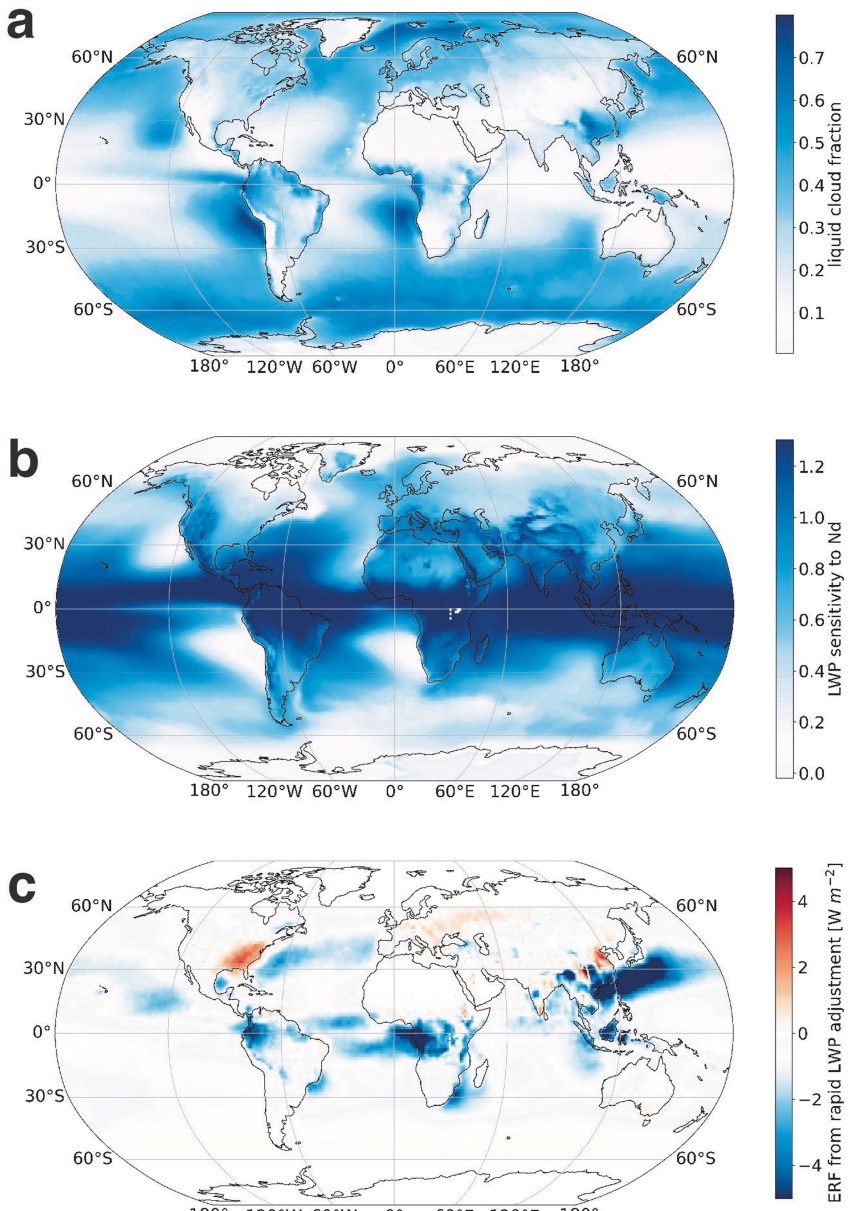

**Extended Data Fig. 6 | The resulting forcing from LWP adjustments is concentrated in low-stability, high cloud fraction regions.** Shown are spatial patterns of quantities related to radiative forcing. a) Liquid cloud retrieval fractions from MODIS[41] 2014-2019 are multiplied with b) LWP sensitivities calculated from ship track observations, extrapolated globally according to EIS, as well as the downwelling shortwave radiation from ERA5[44] and the estimate for anthropogenic $N_d$ changes from Bellouin et al[8]. This gives an estimate c) of radiative forcing from the adjustment of LWP to anthropogenic $N_d$ changes (plotted is the case of higher cloud fraction and anthropogenic $N_d$ enhancement). Maps from cartopy[42].

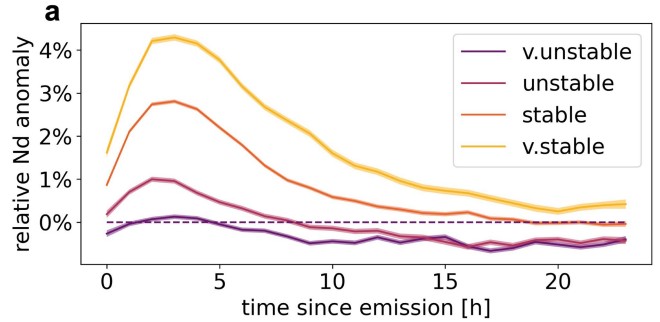

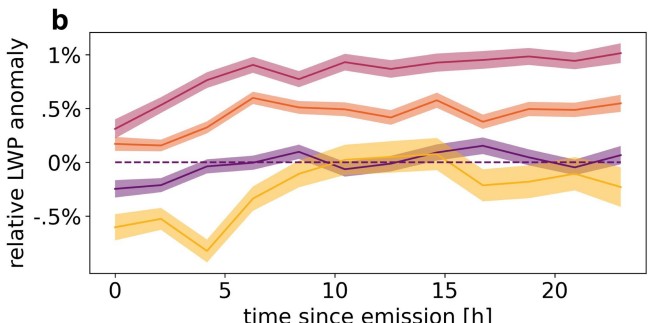

**Extended Data Fig. 7 | Comparing $N_d$ and LWP anomalies under different stability conditions, as measured by EIS.** Plotted are ratios of hourly means of in-track and out-of track properties, so that a value larger than 0% means an enhancement in the track. Shaded areas show standard errors of the means.

LWP anomalies are given in 2h means rather than 1 h because of noisier data. The boundaries of EIS bins are given in Extended Data Table 1. Retrievals from MODIS-cloud product on Aqua and Terra[41], all of 2014-2019.

**Extended Data Table 1 | Sensitivity values $\beta_L$ obtained for different bins of EIS, as well as uncertainties**

| EIS bounds [K] | $\beta_{\mathcal{L}}(EIS)$ | $\pm$ |
|---|---|---|
| -12.7 to 0.5 | nan (5.1) | nan (5.6) |
| 0.5 to 3.5, | 1.30 | 0.02 |
| 3.5 to 9.0 | 0.21 | 0.01 |
| 9.0 to 33.8 | -0.03 | 0.01 |

In the lowest EIS bin, sensitivities are not reliably available as there is no $N_d$ enhancement.