## [Peer Review File · Nature]

Manuscript Title: Invisible Ship Tracks Show Large Cloud Sensitivity to Aerosol.

Reviewer Comments & Author Rebuttals

Reviewer Reports on the Initial Version:

Referees' comments:

Referee #1 (Remarks to the Author):

In this paper, Ship Emissions are estimated at the level of individual ships following the bottom-up methodology outlined in the STEAM2 model (2012) based on the power delivered by ship's engines (Main and Auxiliary) for each speed and operation mode. STEAM2 combines the ship activity data from AIS with technical data about the ship type, propulsion, and auxiliary power requirements based on the IHS Fairplay database. This marks substantial progress on the existing other global inventories before 2012 but, there are certain parameters that that are not well defined and should be taken into account as will be seen below.

STEAM2 model is based on the relationship of the instantaneous speed to the design speed in a way that propulsion engine operating power varies with the cube of vessel speed (propeller law) and the speed safety margin. But the relationship of the instantaneous speed to the design speed is not a cubic because the exponent depends on the type of ship and varies from 3 (Bulk Carrier) to 3.8 (Ro-pax ships) (MAN energy solutions). The effect of wind and waves is also included in the STEAM2 model but does not specify when the ship is in coastal navigation and ocean navigation, the effects on the power that the engines of the ship must develop are not the same. On the other hand , the Main Engine loads during voyages are determined based on the ratio of ship speed and the calculated resistance that the ship is required to overcome at a specified speed. But it also depends on the fouling (hull and propeller), the effect of current on the speed, the effective effects on the energy consumption(Power) by trim and the efficiency of the propulsion system when the ship is propelled by waterjet sytem (domestic traffic).

All of the above assumes an additional power consumption that is not well defined in STEAM2 model. Emissions could increase by up to 30%

Under my point of view the paper should not be published until all the above parameters are defined so that it is easy for the reader to understand.

Referee #2 (Remarks to the Author):

This paper addresses a key gap in the aerosol-cloud-climate literature by quantifying cloud changes due to ship tracks that are not easily detected via manual methods. In so doing, it identifies an important sampling bias — detectable ship tracks tend to have lower liquid water paths than surrounding clouds, but undetected tracks tend to have greater liquid water paths. This has potentially large implications for aerosol radiative forcing (more negative than currently thought) and therefore climate sensitivity (more positive). The study is overall excellent and I would be excited to see it published in Nature after minor revisions. Congratulations to the authors on their nice work. -MD

General comments:

A. It would helpful if you could engage with the existing modeling literature on aerosol effects in trade cumulus clouds. In particular I was thinking of Seifert et al. (2015), which has some similar

results (cloud deepening in response to higher N_d) but also some contrary results (it argues aerosol should cause trade wind cumulus to have small LWP changes but lower cloud fraction overall, causing negligible forcing).

B. The deepening mechanism in trade cumulus clouds (Fig. 5) makes sense, but could you not show direct evidence for it from cloud top temperature? That analysis may be worth doing and could strengthen the findings.

C. It would be worth engaging more with the time dependence argument in Glassmeier et al. (2021). For example, my first reaction to the LWP results for visible tracks/in Sc regions in Figs. 2-3 was that it contradicted the G+21 argument for a strengthening negative LWP response over time. However, if you were to plot $d\ln(\text{LWP})/d\ln(N_d)$ over time I think the results would be consistent. That type of plot could be a worthwhile addition in the Extended Data.

Specific comments:

1. Figure 1: Is it right to say this is "false colours"? I associate that with something like combining three MODIS bands that aren't all in the visible (e.g., to see cloud phase more clearly). This is just using a color scale to indicate cloud droplet effective radius, which is a different concept. You *could* use a false color image to look at ship tracks (e.g., many of Velle Toll's figures) but that doesn't appear to be what you're doing here.

2. Lines 121-126; Figure E1: Would it be worth adding background N_d to this plot? It seems like a relevant consideration along with the meteorological variables you already show.

3. Line 124: "Stronger inversion" might be clearer than "more stable" here, just to clarify you're discussing stability in terms of capping of the MBL and not stability within the MBL (decoupled MBLs are not likely to have more visible ship tracks).

4. Lines 151-153: For a perfectly adiabatic cloud at a fixed cloud base, to increase LWP, clouds would need to deepen. But real clouds are subadiabatic, and changes in how far they are from perfect adiabaticity can change LWP. Also, changes in cloud base can contribute to cloud thickness changes, not just in cloud top.

5. Line 155: I don't think you've introduced EIS? Could be good to do when you introduce Figure E1 (see comment about "stability" above).

6. Figure 4: What is happening in the Caribbean and Mediterranean in terms of relative N_d anomalies? Is it really negative in ship tracks, or is that likely noise/error? Is there a way you could quantify statistical significance of the anomalies here, like via two standard deviations (as in Fig. E3)? If those are significant results, I'm puzzled and a bit concerned about how the method is working in those regions.

7. Lines 186-194: I think this discussion is mostly right, but have a few nuances the authors may wish to consider in comparing to my 2020 paper and Seidel et al. (2014). First, those studies both measure regional top-of-atmosphere albedo directly, not just cloud albedo. Your method cannot distinguish whether the presence of the ship tracks actually cause decreases in cloudiness elsewhere (ala Wang et al., 2011, ACP) which would offset the overall regional albedo effect. None of the three studies can speak to the possibility of remote regional offsets (e.g., via teleconnections), except for the global results in Seidel et al. Just worth keeping in mind for relevance to MCB. Second, you aren't measuring albedo directly, but rather a proxy based on cloud optical thickness. This proxy is quite good so perhaps this shouldn't matter much, but it does make me take the result here with a few more grains of salt than I would a direct measurement. Finally, cloud fraction changes aren't addressed here, but would of course be relevant for regional

TOA albedo and were accounted for in Diamond et al. (2020), although not specifically in the cloud albedo "time of emergence" figure in question.

8. Line 193: You may want to define what "confidently" means here, as I originally interpreted it as ">95% confidence" but from Fig. E3 it seems closer to "more likely than not".

9. Lines 200-201: This may be technically true for ship track studies per se, but Diamond et al. (2020) and the series of papers by Karsten Peters and coauthors did look at trade wind regions on the regional/shipping lane scale. Of course, we weren't able to find statistically significant results (with the partial exception of Nd in my paper), so the present work is very novel in establishing clear effects here. It's interesting to note that the mean LWP result in Diamond et al. (2020) in the tropical (trade wind) regions is qualitatively consistent with your results. I thought that was likely noise and thus meaningless, but now I have a good reason to reexamine the data!

10. Figure E4: Is this technically liquid cloud retrieval fraction?

References:

Peters, K., Quaas, J. & Graßl, H. A search for large-scale effects of ship emissions on clouds and radiation in satellite data. *Journal of Geophysical Research: Atmospheres* 116, D24205, doi:10.1029/2011jd016531 (2011).

Peters, K., Quaas, J., Stier, P. & Graßl, H. Processes limiting the emergence of detectable aerosol indirect effects on tropical warm clouds in global aerosol-climate model and satellite data. *Tellus B: Chemical and Physical Meteorology* 66, 24054, doi:10.3402/tellusb.v66.24054 (2014).

Seifert, A., Heus, T., Pincus, R. & Stevens, B. Large-eddy simulation of the transient and near-equilibrium behavior of precipitating shallow convection. *Journal of Advances in Modeling Earth Systems* 7, 1918-1937, doi:10.1002/2015ms000489 (2015).

Wang, H., Rasch, P. J. & Feingold, G. Manipulating marine stratocumulus cloud amount and albedo: a process-modelling study of aerosol-cloud-precipitation interactions in response to injection of cloud condensation nuclei. *Atmospheric Chemistry and Physics* 11, 4237-4249, doi:10.5194/acp-11-4237-2011 (2011).

Referee #3 (Remarks to the Author):

Review of

"Invisible Ship Tracks Show Large Cloud Sensitivity to Aerosol",
Peter Manshausen, Duncan Watson-Parris, Matthew Christensen, Jukka-Pekka Jalkanen, and Philip Stier

This review only evaluates the use of HYSPLIT in the analysis, as stipulated by the Editor, and as is appropriate based on this reviewer's particular expertise.

The authors have used HYSPLIT to advect ship emissions forward in time in order to analyze their potential cloud-influencing and other impacts with satellite observations at the times/locations of downwind satellite overflights. In this way, the analysis has been able to consider ship-track emissions impacts even if impacts cannot be "seen" from satellite overflights.

Overall, the use of HYSPLIT in this analysis seems appropriate and reasonable. The use of ERA5 reanalysis data with 3-hour data temporal resolution and 0.25 degree horizontal grid resolution is

commended, and it is recognized that additional effort was expended to convert the ERA5 data to HYSPLIT format in order to use these data to drive the HYSPLIT model. While converters are available in the HYSPLIT modeling suite, they are not trivial to use. It is noted that HYSPLIT interpolates the driving meteorological data temporally and spatially, and so estimated air mass pathways are not as "crudely" estimated as might be surmised from the 3-hr, 0.25 deg meteorological data resolution.

Based on the context and description of the analysis, it appears that the trajectory functionality of the HYSPLIT model was used, as opposed to dispersion simulations, but it would be helpful if the authors stated that fact in the methodology section. Trajectories represent the centerline of an advected plume. It is possible that an analysis using a dispersion simulation, instead of a trajectories, could have been used. But, given the inherent uncertainties, I would not expect that any significant advantage would have been afforded by introducing this additional complexity into the analysis. To this point, however, it would probably be useful for the authors to mention that the inherent uncertainties the ERA5 met data -- as would be the case with any met data used to drive HYSPLIT -- does introduce some unavoidable uncertainty into the analysis. This is not a criticism of the analysis, but only a suggestion that the authors acknowledge this source of uncertainty. Meteorological model analyses over the oceans tends to be more uncertain than over many terrestrial regions because there are less met data observations to initialize and assimilate into the numerical weather prediction model.

The use of a 20 meter starting height seems appropriate, and the use of "model vertical motion" also seems appropriate in this application of the HYSPLIT model.

In summary, I believe that the use of the HYSPLIT model in this analysis appears to have been appropriate.

Author Rebuttals to Initial Comments:

We thank all reviewers for their thoughtful and constructive feedback that helped us to further advance the manuscript. We are pleased that you found our work ‘overall excellent’ and our methodology ‘appropriate and reasonable’, as well as pointing out that it ‘addresses a key gap in the aerosol-cloud-climate literature’. We address individual comments below.

Reviewer 1

1 ...All of the above assumes an additional power consumption that is not well defined in STEAM2 model. Emissions could increase by up to 30%.

We acknowledge that there may be uncertainties of the emission amounts stemming from the STEAM model. However, in this work, we do not use the emission amounts, but only the locations of emissions. Any uncertainty in emission amounts will therefore not affect the results presented here, nor the conclusions of this study. We have added a clarification about this in the Methods section. Even though not directly relevant to this study, we include a more in-depth discussion of the STEAM model and related uncertainties in a separate document.

- l. 477ff: Added: “*Note, that while we calculate emission magnitudes, the analysis in this study uses only the emission locations. While there are uncertainties in the emission amounts, these do not alter the findings of the study.*”

Reviewer 2

2A. It would helpful if you could engage with the existing modelling literature. I was thinking of Seifert et al. (2015), which has some similar results (cloud deepening in response to higher Nd) but also some contrary results (it argues aerosol should cause trade wind cumulus to have small LWP changes but lower cloud fraction overall, causing negligible forcing).

We welcome the suggestion to compare more with modelling results, and have included a comparison with Seifert et al. (2015), who provide a detailed description of modelled shallow cloud response to Nd variations. Similar results were obtained by Yamaguchi et al. (2019), focusing on the role of wind shear, which we have also now included. We also already comment on the studies by Dagan et al. (2015), which proposes ‘optimal concentrations’ of Nd as well as Spill et al. (2019), which argues that the transient response should be considered rather than the equilibrium one and matches our observations of LWP increases more closely. Lastly, we also mention the results of Toll et al. (2019) and Possner et al. (2020), which are at odds with our results of increased LWP in regions of weak inversions (low EIS, deep boundary layers).

- l. 186ff: added: “*An LES study of trade cumulus clouds by Seifert et al. (32) found cloud deepening with increasing Nd, but only small LWP increases. However, while Seifert et al. focus on the equilibrium states of their simulations, Spill et al. (33) find a more robust positive LWP response and cloud deepening in simulations of more realistic, transient thermodynamic conditions. Yamaguchi et al. (34) also find deepening and LWP increases in trade cumulus, but only conditional on no-wind-shear conditions.*”
- l. 174ff: added: “*Lower EIS is correlated to deeper boundary layers, so the increase in LWP in low-EIS regions is contrary to results from visible track observations by Toll et al. (31) as well as simulations by Possner et al. (32), who find a more negative LWP response in deeper boundary layers.*”

2B. The deepening mechanism in trade cumulus clouds (Fig. 5) makes sense, but could you not show direct evidence for it from cloud top temperature?

This is a really good point. On the back of the envelope, we expect $LWP \sim CTH^{**2}$; $CTH \sim CTT$; and so if LWP increases by 2%, that means CTH increases by $\sqrt{1.02} \sim 1.01$, so from e.g. 1000 to 1010. With 100m \sim 1K this means a 0.1K decrease. This is consistent with the changes plotted in Fig. E4, but very small compared to the noise in the time-resolved case. But it would be interesting to investigate this further with another instrument that does direct measurements, like CALIOP in Christensen et al. (2011), although we note that the vertical resolution of CALIOP is also limited to around 30-60m.

- added: *Fig. E4,*

- l. 163ff: added: “Fig. E 4 shows the anomaly in cloud top temperature (CTT), which decreases with cloud height. Regions with stronger LWP responses seem to show cloud top increases (temperature decreases). However, the signal is not strong enough to allow for a significant time-resolved study.”
- l. 249ff: changed to: “Retrieval uncertainties also need to be considered: LWP retrievals in the broken cloud scenes of the trade wind Cu are much more uncertain than in Sc decks (39). CTT retrievals suffer from uncertainties under strong-inversion conditions.”

2C. It would be worth engaging more with the time dependence argument in Glassmeier et al. (2021). For example, my first reaction to the LWP results for visible tracks/in Sc regions in Figs. 2-3 was that it contradicted the G+21 argument for a strengthening negative LWP response over time. However, if you were to plot $\ln(LWP)/\ln(Nd)$ over time I think the results would be consistent.

Good idea! We added this plot (E1). While on the face of it, the Figure we add supports it, we would caution against seeing this result as evidence for the argument advanced by Glassmeier et al.: the reason for the sensitivity increase is the time-development of Nd response. The LWP response, looking at Fig. E7, actually is getting more positive with time, rather than more negative. (Here, all cloud regimes/regions are taken together.) Errors also increase with time in E1, as the Nd signal becomes weaker relative to the noise.

- added: Fig. E1
- l. 119ff: added: “For the resulting sensitivity of LWP to Nd perturbations, defined in equation (1), this means a negative sensitivity for visible tracks and a positive sensitivity for invisible tracks. The time-development is plotted in Fig. E 1. In agreement with Glassmeier et al. 19, the magnitude of sensitivity increases. Here, however, this seems to be due to the decreasing Nd anomaly with time, rather than an increasing LWP anomaly.”

2.1. Figure 1: Is it right to say this is "false colours"?

You are right, this is indeed just a color scale, not a composite and false colors.

- Changed to: “A view of the Chilean stratocumulus deck from the MODIS satellite instrument, with the color scale...”

2.2. Lines 121-126; Figure E1: Would it be worth adding background Nd to this plot?

Yes, this is a good idea!

- (now Fig. E2): added background Nd, which is more peaked at lower Nd for visible cases
- l. 130f: added: “and have a more peaked Nd distribution.”

2.3. Line 124: "Stronger inversion" might be clearer than "more stable" here

Agreed!

- now l. 128ff: changed to: “capped by stronger inversions (as quantified by Estimated Inversion Strength, EIS), ...”

2.4. Lines 151-153: For a perfectly adiabatic cloud at a fixed cloud base, to increase LWP, clouds would need to deepen. But real clouds are subadiabatic, and changes in how far they are from perfect adiabaticity can change LWP. Also, changes in cloud base can contribute to cloud thickness changes, not just in cloud top.

We acknowledge the possibility of subadiabatic clouds, for example in connection to entrainment at cloud top. Clouds could indeed also deepen ‘downwards’. We include a reference to this as below. However, both of these would not help to explain the observed differences between strong and weak inversion regions so we believe our original argument is likely to dominate.

- l. 162: changed to: “However, cloud deepening upwards...”
- l. 167f: added: “At the same time, changes in adiabaticity or deepening downwards could also increase LWP.”

2.5. Line 155: I don't think you've introduced EIS?

Apologies, we have now included it, as per comment 2.3.

2.6. Figure 4: What is happening in the Caribbean and Mediterranean in terms of relative Nd anomalies? Is it really negative in ship tracks, or is that likely noise/error?

This is a negative signal, but it is likely to arise from the way the counterfactual is calculated and can happen in the case of strong background aerosol gradients (see the change below). See also Figure R1 included below, showing that indeed, there is a strong background Nd gradient in the Caribbean. Plotted along longitude, the background is a convex function in each box, explaining the (unphysical) negative response in Nd in the Caribbean. We partially address this issue by creating counterfactuals on either side of the tracks but this effect will likely be present to some extent everywhere on the map, with a changing sign. Averaging everywhere, it should disappear.

- l. 508ff: added: “Some error may be introduced by strong background aerosol gradients, especially when the distributions are nonlinear. Then, the counterfactual constructed by averaging the areas to either side may be an overestimate (underestimate) for a convex (concave) function in the across-track direction. This explains the unphysical Nd decrease in tracks, e.g. in the Caribbean. Over large enough, backgrounds should on average not be convex or concave, so that averaging over the entire observation region, these effects should be negligible.”

Figure R1: Background Nd concentrations in the Caribbean, for the two boxes that show strong negative Nd responses in Fig. 4. Plotted is data for years 2017-2019

2.7. Lines 186-194: Diamond et al. (2020) and Seidel et al. (2014): First, those studies both measure regional top-of-atmosphere albedo directly, not just cloud albedo. Your method cannot distinguish whether the presence of the ship tracks actually cause decreases in cloudiness elsewhere (ala Wang et al., 2011, ACP) which would offset the overall regional albedo effect. [...] Second, you aren't measuring albedo directly, but rather a proxy based on cloud optical thickness. This proxy is quite good so perhaps this shouldn't matter much, but it does make me take the result here with a few more grains of salt than I would a direct measurement. Finally, cloud fraction changes aren't addressed here, but would of course be relevant for regional TOA albedo and were accounted for in Diamond et al. (2020), although not specifically in the cloud albedo "time of emergence" figure in question.

We agree with the points raised here that the application to MCB experiments comes with some caveats, notably adjustments away from the track, the albedo proxy we use, and cloud fraction changes. We still think that these could be addressed in the framework of a targeted observation, and see our method as a possible step in this direction.

- l. 216ff: added: “For the purpose of MCB experiments, there is the caveat that we do not consider cloud fraction changes or possible decreases in cloudiness at larger distances from the tracks as found by Wang et al. (37). Furthermore, our albedo measurements rely on the proxy of cloud optical thickness, similarly to other studies of this kind (23, 15).”

2.8. Line 193: You may want to define what "confidently" means here.

Yes, good point, the Figure was somewhat misleading. We extended it to the four months described in the text, at which point we can detect an albedo anomaly at 95% significance in 95% of the random draws of days. We also clarify in the legend.

- Fig. E5: extended to 4 months, reduced the number of draws to balance increased computing requirements due to extension.
- added (in caption): *“We assume that we are able to make a confident detection if in 95% of draws, so 19 out of 20, we observe significant anomalies.”*

2.9. Lines 200-201: This may be technically true for ship track studies per se, but Diamond et al. (2020) and the series of papers by Karsten Peters and coauthors did look at trade wind regions on the regional/shipping lane scale. Of course, we weren't able to find statistically significant results (with the partial exception of Nd in my paper), so the present work is very novel in establishing clear effects here.

Yes, on the scale that does not follow individual tracks, as we do here, the trade wind region has been studied – as you say, not finding significant changes except for Nd. We now mention these studies as below, and also acknowledge them in the discussion.

- l. 145ff: Added: *“Previous studies on the shipping lane or regional scale (21,26,27) in trade cumulus have not found statistically significant anomalies, with the exception of Nd enhancements (21).”*
- l. 226f: added: *(rather than regional shipping effects (21,26,27))”*

Reviewer 3

3.1. The use of ERA5 reanalysis data with 3-hour data temporal resolution and 0.25 degree horizontal grid resolution is commended, and it is recognized that additional effort was expended to convert the ERA5 data to HYSPLIT format in order to use these data to drive the HYSPLIT model.

Thank you for this. We took this opportunity to point out this contribution by M.C.

- l. 598: added: *“M.C. converted the ERA5 data for use with the HYSPLIT model.”*

3.2. It is noted that HYSPLIT interpolates the driving meteorological data temporally and spatially, and so estimated air mass pathways are not as "crudely" estimated as might be surmised from the 3-hr, 0.25 deg meteorological data resolution.

Indeed, and it is a good idea to mention this explicitly.

- l. 489f: added: *“Note, that ERA5 data comes at 3h and 0.25 degree resolution, but that the HYSPLIT model interpolates for a more exact estimate.”*

3.3. ... it appears that the trajectory functionality of the HYSPLIT model was used, as opposed to dispersion simulations, but it would be helpful if the authors stated that fact in the methodology section.

Agreed, we add this to the Methods and also explain, that it would be computationally too expensive to use the dispersion mode. This is because it creates a 2D object from a point (source) which means orders of magnitude more data. We agree that this should not introduce significant uncertainty given the uncertainties of the meteorological data used to drive the model.

- l. 492ff: added: *“The HYSPLIT functionality for trajectories, rather than that for dispersion is used here, so as to find the advected emission location. This is to limit computational cost, as the final data sets analyzed are large.”*

3.4. It would probably be useful for the authors to mention that the inherent uncertainties the ERA5 met data -- as would be the case with any met data used to drive HYSPLIT -- does introduce some unavoidable uncertainty into the analysis.

Yes, this is a very good point and we have added this to the discussion as a source of uncertainty. As with the other points, this could affect the reported Nd and LWP anomalies, but it should not impact the calculated LWP/Nd sensitivity, because it would affect both measurements in the same way.

- I. 242f: added: *“The origin of this is the inherent uncertainty of the reanalysis data used to advect the track, uncertainties related to trajectory modelling, as well as uncertainty of the original emission location.”*

Description of STEAM calculation process

The reviewer raised some concern over the assumptions and uncertainties involved in ship emission model STEAM.

Note, however, that the current manuscript uses STEAM SO_x emission output only to pinpoint the location and time of ship plumes. In other words, the STEAM output is used to determine whether at a specific location and at the given time, there should exist a ship plume or not. We do not quantify the difference between predicted emissions of SO_x and the corresponding signal from observations. The comparison of the magnitude of the effect between emission modeling and observations are left for future work. In that sense, the uncertainty issues of STEAM raised by the reviewer are not directly relevant to this paper.

However, we would like to clarify some misconceptions which may arise from use of STEAM version which was released more than ten years ago. A description below outlines the datasets and the calculation process of STEAM in an introductory manner, to provide an overview of the algorithms included in the modeling process. This is not intended as an exhaustive description, but a short summary to clarify some of the features included in the emission modeling which go beyond the propeller law.

Input data

The Ship Traffic Emission Assessment Model of the Finnish Meteorological Institute uses as input the following datasets.

- a) *Vessel activity*. Most of the time this consists of global Automatic Identification System (AIS) transponder messages, which include data both from terrestrial and satellite AIS networks. However, also Long-Range Identification and Tracking (LRIT) data, Vessel Monitoring System (VMS) and vessel arrival/departure times can be used. In principle, anything with a UTC time label, vessel identity and vessel position can be used. Instantaneous speed may or may not be included but can be used by the model if available. Global vessel activity datasets are provided by commercial operators and restricted to FMI research purposes. These are currently provided by Orbcomm Ltd.
- b) *Global fleet description*. These data include technical features of all the ships in the global fleet and are provided by a commercial operator. Required data include physical dimensions, machinery, propulsion system details, power generation and transmission features, capacity description and installation of emission abatement techniques (Sewage treatment plants, catalytic converters, exhaust gas cleaning systems, exhaust gas recirculation units, ballast water management systems etc). In this case, data from IHS Markit and IMO GISIS are used.
- c) *Environmental conditions*. These datasets describe the ambient conditions in which vessels operate. These are usually provided by Copernicus Marine Environment Monitoring Systems (CMEMS) and Copernicus Atmospheric Monitoring Services, and they consist of gridded spatiotemporal global datasets of sea ice coverage, significant wave heights, wave directions, sea current speed/direction, surface wind speed/direction. Environmental contributions to emissions are turned off by default but can be included if necessary.
- d) *Polygon descriptions of special areas*. These have been drawn based on international, regional, and local regulations concerning various pollution streams from shipping. For example, IMO MARPOL Annex VI defines Emission Control Areas (ECAs) for air emissions (SO_x ECA, NO_x ECA); additional EU rules for sulfur emissions and national rules for Chinese

domestic ECAs have been implemented and starting dates defined for each region. For ship discharges, other international conventions are used (MARPOL I, IV, AFC, BWMC).

These input datasets define, for each vessel, its capabilities of using various fuels during the modeling runs. These are defined by engine properties, operation area and time stamp. Technical capability alone cannot be used in determination of fuel type and properties because also date of entry to force of various rules need to be considered.

A simplistic description of the STEAM calculation process is given in Figure 1. The details of the calculation process are given elsewhere (Jalkanen et al., 2012; Johansson et al., 2017) for air emissions, discharges (Jalkanen et al., 2021) and noise (Jalkanen et al., 2018). This document concentrates on air emissions modeling description and especially on the power prediction methodology in STEAM.

Figure 1 Conceptual schema of STEAM calculation process (Johansson et al., 2017)

Performance prediction

Most of the ship emission models in available literature use directly a simplistic cubic relation of speed to design speed (Comer et al., 2017; Faber et al., 2020; Schwarzkopf et al., 2021; Smith et al., 2015) as mentioned by the reviewer. This was also the starting point of STEAM in 2009 (Jalkanen et al., 2009), but the performance prediction algorithm was updated to the Hollenbach resistance prediction method in 2012 (Jalkanen et al., 2012). The Hollenbach method description is a parameterized model based on resistance tests of 433 vessels and considers e.g. vessel hull shape, bulbous bows, and different resistance components. This means that the simple cubic relationship between speed to design speed is not applied in STEAM anymore.

In STEAM, vessel resistance is determined by the following components.

$$R_{\text{total}} = R_{\text{residual}} + R_{\text{friction}} + R_{\text{fouling}} + R_{\text{air}} + R_{\text{ice}}$$

Once R_{total} is known, the necessary engine power is determined from

$$\text{MEPower} = R_{\text{total}} * v_{\text{inst}} / \text{qpc},$$

in which the qpc is the quasi-propulsive constant which includes the propulsive losses of power transmission and propeller efficiency. The qpc calculation is described by Watson (Watson, 1998) and includes contributions from propeller rotation speed and vessel length. Additional components to performance prediction include also the effect of waves and sea current, but these are considered as modifications of vessel speed (knots), not resistance (kilonewtons).

The individual resistance components of R_{tot} are modeled using the following methods.

R_{residual} comes from the Hollenbach resistance prediction

R_{friction} uses the ITTC method to determine hull friction (ITTC – Recommended Procedures and Guidelines Resistance and Propulsion Test and Performance Prediction with Skin Frictional Drag Reduction Techniques, 2017).

R_{fouling} uses the ITTC formulation for hull surface roughness. Note, that a proper description of the fouling contribution would require knowledge of drydocking/hull cleaning activities, water temperature, salinity, availability of sunlight and so on, to properly account for the impact of soft/hard fouling to vessel resistance. This part is currently done in a simplistic manner using the delta_cf term mentioned in the ITTC documentation. The user has the option to override this behavior and provide a percentage value instead. This override mechanism can be used to make the fouling contribution identical to the assumption used in the IMO GHG studies.

R_{air} uses the Blenderman algorithm (Blendermann W: Wind loading of ships collected data from wind tunnel tests in uniform flow, 1996; see Bertram and Schneekluth, 1998 for a summary) to describe the effect of wind to vessel resistance. This is a function of ship superstructure cross-section, wind speed and wind angle. The wind velocity and direction data are obtained from Copernicus services (Figure 2).

Figure 2 Example of 10m wind data (U component) used in STEAM modeling. This data is taken from Copernicus Atmospheric Monitoring Services of the ECMWF.

Rice is described considering the Finnish-Swedish ice class rules using the method proposed by Riska (Riska et al., 1997), which considers certain hull angles, bollard pull features and operational modes (independent icebreaking, assisted by icebreaker, towing, sailing along broken ice channel). Data are taken from CMEMS ice thickness data product (Figure 3).

Sea ice thickness

Figure 3 An example of a global sea ice thickness map included in STEAM calculations. These data were obtained from Copernicus Marine Environment Monitoring Services of the ECMWF.

The residual resistance term includes (Hollenbach, 1998):

$$R_{\text{residual}} = C_r \cdot \rho \cdot \frac{1}{2} \cdot v_{\text{inst}}^2 \cdot (B \cdot D / 10)$$

As can be seen from this equation, sea water density (ρ), vessel breadth (B) and draught (D) are included as well as the v_{inst}^2 term. The final cubic term is obtained when R_{total} is multiplied with the v_{inst} .

The C_r is a resistance coefficient which is determined from $C_{r_{\text{Fncrit}}}$, L_{os} , B , D_{aft} , D_{fwd} , D_{midship} , N_{thruster} , N_{rudder} , N_{bossings} , D_{prop} , N_{brackets} , C_b , displacement and various other resistance parameters found in the original reference (Hollenbach, 1998) or in a book from Bertram and Schneekluth (Bertram and Schneekluth, 1998). The C_r equation has contributions to vessel dimensions (Length over surface, L_{os}), trim (draught at midships, aft and bow, D_{midship} , D_{aft} , D_{fwd}), propeller features (propeller diameter, D_{prop}), hull form (Block coefficient, C_b) and displacement, which usually is enough to describe the wet surface area and residual resistance components quite accurately. In the Hollenbach method, impact of trim on vessel performance can be included, but this information is not available for the global fleet as a function of time, and vessel trim is not considered in our modeling. From technical point of view, inclusion of trim is possible in STEAM, but the unavailability of vessel draught data sets a restriction in this regard. For vessel draught, design draught is used by default, because the draught value included in AIS data is not present or updated for majority of vessels seen in AIS data.

Note, that if ambient contributions to vessel resistance are not considered, R_{air} , R_{ice} , are zeros and can be neglected. If the effect of sea currents and waves are not included, then their contribution to vessel speed changes are set to zero. Wave and sea current components directly modify the vessel speed to obtain speed over water (modifies the speed entry of AIS position report, because speed over ground in AIS is not usually equal to speed over water).

The effect of waves (Figure 4) is included with the same method already introduced in the first version of STEAM (Jalkanen et al., 2009) which is based on the method developed by Townsin (Townsin et al., 1993) and provides $\Delta v/v$ which introduces a speed penalty caused by waves (in percent). Vector math is applied for impact of sea current (direction, speed of water flow) to vessel speed (Figure 5).

Significant wave height

Figure 4 An example of significant wave height data used in calculation of the effect of waves to vessel power prediction. These data were obtained from Copernicus Marine Environment Monitoring Services of the ECMWF.

Eastward Current Velocity

Figure 5 An example of sea surface current data used in STEAM power prediction. Data from Copernicus Marine Environment Monitoring Service are obtained from ECMWF.

We are not going to go further into details with this introduction of STEAM calculation example but provide this introduction to exemplify the process which is repeated for all the ships and every position report in the global AIS dataset. Detailed description of each ambient contribution will be provided in an upcoming paper, which is in writing.

This example shows that the performance prediction in STEAM goes well beyond the simplistic cubic v_{inst}/v_{design} relationship mentioned by the reviewer. Indeed, it should be noted that most of the ship emissions models work exactly as the reviewer indicated, applying the cubic v_{inst}/v_{inst} law, but STEAM does the performance prediction in a different way. Although, the cubic dependency is present also in the Hollenbach step, it should be noted that the additional contributions from ambient conditions will modify this dependency. This may lead to situation raised by the reviewer where the simple cubic dependency does not necessarily apply anymore. However, we would like to point out that neglecting the ambient contributions still provides a reasonable accuracy in predicted fuel consumption and associated air emissions.

In fact, the scaling factors for “weather” and “fouling” are far too simplistic (like those applied in the Third and Fourth IMO GHG study) because they do not consider actual weather data, but instead use predetermined constants to scale the results. Whether the vessel is close to the shoreline or not does not matter because, in our approach, the global gridded data from relevant atmospheric/marine datasets are used to describe the ambient contributions to vessel performance prediction. This also removes the artificial need for definitions “close to shoreline” and “open sea” used in the IMO GHG4 study.

Uncertainty of STEAM predictions

The reviewer pointed out potential problems in the accuracy of STEAM arising from different assumptions. We have made an extensive evaluation of STEAM against the EU Monitoring, Reporting and Verification fuel reporting, one such example can be found in the HELCOM Maritime20 meeting document (Maritime20/5-2.INF, available from <https://portal.helcom.fi/meetings/MARITIME%2020-2020-787/MeetingDocuments/5-2%20Emissions%20from%20Baltic%20Sea%20shipping%20in%202006%20-%202019.pdf>), Chapter 3.

As a summary, the overall accuracy of the predicted CO₂ emission inventory was +0.2% overprediction. For individual vessels, both positive and negative uncertainties were found. For any individual vessel, the average absolute deviation was 19% (Figure 6). It should be noted that this comparison did not include any contributions from ambient conditions, but uncertainties concerning data gaps in vessel activity data (AIS blackout), vessel technical description (gaps in IHS Markit data fields) and erroneous MRV reports are all included.

Figure 6 An example of STEAM vs MRV fuel consumption comparison for bulk cargo ships. The colored and numbered points indicate specific vessels with significant gaps in technical description, which lead to larger than normal uncertainty of predicted fuel consumption. The dotted lines indicate 10% error, the solid line indicates a 1:1 fit between STEAM and MRV. The average absolute deviation is 19% for bulk cargo ships. This comparison does not include any consideration for ambient effects.

If all necessary data are available for STEAM, the average absolute error for any individual vessel was determined to be about 10% (without considering the impact of weather).

References

- Bertram, V. and Schneekluth, H.: Ship Design for Efficiency and Economy, 2nd ed., Butterworth-Heinemann, Oxford, UK., 1998.
- Comer, B., Olmer, N., Mao, X., Roy, B. and Dan, R.: Black Carbon Emissions and Fuel Use in Global Shipping, 2015, Int. Counc. Clean Transp., (October), 2017.
- Faber, J., Hanayama, S., Zhang, S., Pereda, P., Comer, B., Hauerhof, E., Schim van der Loeff, W., Smith, T., Zhang, Y., Kosaka, H., Adachi, M., Bonello, J., Galbraith, C., Gong, Z., Hirata, K., Hummels, D., Kleijn, A., Lee, D., Liu, Y., Lucchesi, A., Mao, X., Muraoka, E., Osipova, L., Qian, H., Rutherford, D., Suárez de la Fuente, S., Yuan, H., Velandia Perico, C., Wu, L., Sun, D., Yoo, D. and Xing, H.: The Fourth IMO GHG Study, London, UK., 2020.
- Hollenbach, K. U.: Estimating resistance and propulsion for singlescrew and twin screw ships, Sh. Technol. Res., 45(2), 72–76, 1998.
- Jalkanen, J.-P., Johansson, L., Liefvendahl, M., Bensow, R., Sigray, P., Östberg, M., Karasalo, I., Andersson, M., Peltonen, H. and Pajala, J.: Modeling of ships as a source of underwater noise, , (April), 1–18, 2018.
- Jalkanen, J.-P., Johansson, L., Wilewska-Bien, M., Granhag, L., Ytreberg, E., Eriksson, K. M., Yngsell, D., Hassellöv, I.-M., Magnusson, K., Raudsepp, U., Maljutenko, I., Winnes, H. and Moldanova, J.: Modelling of discharges from Baltic Sea shipping, Ocean Sci., 17(3), 699–728, doi:10.5194/os-17-699-

2021, 2021.

Jalkanen, J. P., Brink, A., Kalli, J., Pettersson, H., Kukkonen, J. and Stipa, T.: A modelling system for the exhaust emissions of marine traffic and its application in the Baltic Sea area, *Atmos. Chem. Phys.*, 9, 9209–9223, doi:10.5194/acpd-9-15339-2009, 2009.

Jalkanen, J. P., Johansson, L., Kukkonen, J., Brink, A., Kalli, J. and Stipa, T.: Extension of an assessment model of ship traffic exhaust emissions for particulate matter and carbon monoxide, *Atmos. Chem. Phys.*, 12(5), 2641–2659, doi:10.5194/acp-12-2641-2012, 2012.

Johansson, L., Jalkanen, J.-P. and Kukkonen, J.: A comprehensive modelling approach for the assessment of global shipping emissions, in *Air Pollution Modeling and its Application {XXV}*, pp. 367–373, Springer International Publishing., 2017.

Riska, K., Wilhelmson, M., Englund, K. and Leiviskä, T.: Performance of merchant vessels in ice in the Baltic, *Winter Navigation Board.*, 1997.

Schwarzkopf, D. A., Petrik, R., Matthias, V., Quante, M., Majamäki, E. and Jalkanen, J.-P.: A ship emission modeling system with scenario capabilities, *Atmos. Environ. X*, 12, 100132, 2021.

Smith, T., O’Keeffe, E., Aldous, L., Parker, S., Raucci, C., Trau, M., Anderson, B., Agrawal, A., Ettinger, S., Corbett, J., Winebrake, J., Jalkanen, J.-P., Johansson, L., Ng, S., Hanayama, S., Faber, J., Nelissen, D., Hoen, M. ‘t, Lee, D., Chesworth, S. and Pandey, A.: Third IMO Greenhouse Gas Study, *Int. Marit. Organ.*, 2015.

Townsin, R. L. ., Kwon, Y. J. ., Baree, M. S. . and nd Kim, D. Y. .: Estimating the influence of weather on ship performance, *RINA Trans.*, 135, 191–209, 1993.

Watson, D. G. M.: *Practical Ship Design*, Elsevier Science., 1998.

Reviewer Reports on the First Revision:

Referees' comments:

Referee #1 (Remarks to the Author):

In my opinion, referring only to the emissions model applied in this study and taken into account the paragraph added by the authors(l. 477ff), the manuscript could be published in the current form.

Dr. Juan Moreno-Gutiérrez

Referee #2 (Remarks to the Author):

The authors did a nice job responding to the reviewer comments. I believe the manuscript is in publishable shape. I have some minor comments below for the authors to consider. -MD

1. Line 131: In Figure E2, the Nd for visible tracks seems lower, not just "more peaked", and this may a more relevant consideration to point out.
2. I have a question about your discussion of Possner et al (32) — you mention "simulations", but that analysis is entirely observational if I'm not mistaken. Did you mean to cite a different paper of Anna's, or is "simulations" an error?
3. Line 513: Should "areas" follow the phrase "Over large enough"?

Author Rebuttals to First Revision:

We thank the reviewers again for their feedback. We are pleased they found that we have 'done a nice job' addressing their comments. The specific points raised by reviewer 2 are addressed below.

1. Line 131: In Figure E2, the Nd for visible tracks seems lower, not just "more peaked", and this may a more relevant consideration to point out.

Agreed!

l 125: changed: "*higher relative humidity conditions, and have a more peaked Nd distribution*" to "*higher relative humidity conditions, and show lower background Nd.*"

2. I have a question about your discussion of Possner et al (32) — you mention "simulations", but that analysis is entirely observational if I'm not mistaken. Did you mean to cite a different paper of Anna's, or is "simulations" an error?

Apologies, the paper reference is correct, but it should read 'observations' instead of 'simulations'.

l 165: changed: "*climatological simulations by Possner*" to "*climatological observations by Possner*"

3. Line 513: Should "areas" follow the phrase "Over large enough"?

Absolutely, thank you for spotting this typo.

l 489: added: "*areas*"